# Assessing the Impact of COVID-19 on Subjective Well-Being and Quality of Life in Mexico: Insights from Structural Equation Modeling

Ignacio Alejandro Mendoza-Martínez [1], Edmundo Marroquín-Tovar [1], Jorge Pablo Rivas-Díaz [1], Araceli Durand [1], Gustavo Enrique Sauri-Alpuche [1] and Blanca Rosa Garcia-Rivera [2,*]

1    Facultad de Economía y Negocios, Campus Sur, Universidad Anáhuac, Mexico City 01840, Mexico; alejandro.mendozam@anahuac.mx (I.A.M.-M.); edmundo.marroquin@anahuac.mx (E.M.-T.); jorge.rivas@comunidad.unam.mx (J.P.R.-D.); alicia.durand@anahuac.mx (A.D.); gsauri@anahuac.mx (G.E.S.-A.)
2    Facultad de Ciencias Administrativas y Sociales, Universidad Autónoma de Baja California, Ensenada 22890, BC, Mexico
*    Correspondence: blanca_garcia@uabc.edu.mx; Tel.: +52-646-179-4300

**Abstract:** Amid the backdrop of the COVID-19 pandemic, the living conditions of the population were dramatically altered, with social distancing measures and the looming threat to public health leaving a profound impact on people's lives. This study aims to assess the influence of COVID-19 on subjective well-being and overall quality of life in Mexico. A structural model with latent variables was used. Data were extracted from the National Self-Reported Well-being Survey (SWLS) from October 2020 and January 2021, featuring a robust sample size of 3615 residents from urban areas in Mexico, all aged 18 and above. Findings revealed that around 38% of the variance in overall life satisfaction in October 2020 and January 2021 could be attributed to Personal well-being (0.231), Personal satisfaction (0.320), Satisfaction with the environment (0.076), and Negative emotional states ($-0.116$). In comparison, October 2019 to January 2020 saw a lower 20% explained variance, primarily associated with Personal well-being (0.184), Personal satisfaction (0.270), and Satisfaction with the environment (0.052). Reliability assessments, including Cronbach's Alpha coefficients, Rho_a, and Composite Reliability, all surpassed 0.70 for each subscale. In addition, our study confirmed convergent validity, as the Average Variance Extracted (AVE) consistently exceeded 0.50 across all subscales, while the discriminant coefficient exceeded 0.70.

**Keywords:** COVID-19; subjective well-being; emotional states; satisfaction with the environment; self-satisfaction; SWLS; health economics



## 1. Introduction

General life satisfaction, also referred as well-being, is an important construct that has been assessed in Mexico since 2012 using the Self-Reported Satisfaction with Life Scale (SWLS and BIARE in Spanish) [1], within the framework of the National Consumer Confidence Survey (ENCO) [2]. It is administered consecutively four times a year, considering quarterly intervals, during the months of January, April, July, and October each year. Since 2013, the SWLS has been harmonized with other international organizations of The Organization for Economic Cooperation and Development (OECD) member countries, comprising booklets that assess the following aspects: (1) Eudaimonia or happiness, (2) Affective balance, and (3) Specific domains of satisfaction.

Amidst the upheaval caused by the COVID-19 pandemic, global well-being and quality of life have faced unprecedented challenges. The pandemic's pervasive influence has extended far beyond physical health, deeply impacting individuals' emotional and psychological states worldwide. As nations grapple with the complexities of navigating through

this crisis, understanding its profound ramifications on well-being becomes paramount. In this study, we delve into the unique intersection of Mexico's societal fabric and the pandemic's effects, offering a longitudinal analysis that scrutinizes how both economic and psychological landscapes have evolved over time.

This study brings forth novel contributions that stand at the intersection of methodological innovation, contextual specificity within Mexico, and a comparative analysis of well-being pre and post the pandemic. By employing a longitudinal approach coupled with structural equation modeling, we illuminate the nuanced dynamics of well-being amidst the pandemic's tumultuous waves. Such an endeavor is not only timely but essential, as it fills critical gaps in the existing literature, which often lacks comprehensive longitudinal analyses within the context of COVID-19 [3].

While existing research has explored subjective well-being and its multifaceted dimensions, few studies have ventured into the intricate interplay between objective conditions and individual perceptions during such a transformative global event. Our study bridges this gap by elucidating how subjective evaluations intersect with objective realities, shedding light on the intricate relationship between economic fluctuations and psychological well-being [4–6].

Within the broader landscape of subjective well-being research, our study takes a focused approach, synthesizing key findings that directly inform our research questions and hypotheses. By honing in on the longitudinal evolution of well-being amidst the pandemic, we aim to provide actionable insights that inform policy decisions and societal interventions, particularly in navigating the complex terrain of post-pandemic recovery.

In essence, this study not only expands the theoretical understanding of subjective well-being but also offers practical implications for policymakers and practitioners alike. By unraveling the multifaceted nature of well-being amidst unprecedented global crises, we pave the way for informed strategies that foster resilience and recovery in the face of adversity.

Numerous studies analyze subjective well-being, its dimensions, and characteristics to generate valuable information about how individuals perceive it, which is gaining ground in research. Authors such as [6–12], among others, contribute to conceptualizing the relationships and factors that drive population well-being through social and individual impulses and achievements.

The field of subjective well-being research is currently moving towards a holistic approach, emphasizing a comprehensive sense of well-being [13,14]. It encompasses emotional well-being, a sense of purpose in life, social connections, personal character strengths, physical health, and financial security [15]. Importantly, these aspects are highly esteemed by individuals. In addition, many individuals around the world would argue for the inclusion of supplementary well-being domains, such as spirituality or inner peace [1]. Furthermore, it would shed light on the actual research to explore community well-being, which goes beyond a mere summation of individuals' self-reported well-being [16].

Well-being is a complex topic, as it is understood as a complementary concept of both objective and subjective dimensions. Objective conditions such as income level, employment, job stability, and a low cost of living do not directly generate happiness; however, they do contribute to general life satisfaction [17], hence the need to integrate different subjective evaluations to understand well-being comprehensively. This poses a challenge for its modeling in modern society [18,19].

Research argues that subjective well-being encompasses different evaluations that people make about their lives, events, and circumstances, whether positive or negative. Authors like [20–23] defined subjective well-being as the affective cognitive self-assessment associated with an individual's internal state, with indicators like life satisfaction and experienced happiness [24,25].

Previous researchers, such as [26–28], explore the construction of subjective well-being as a multidimensional structure that includes life satisfaction, positive affect, and negative affect. The theoretical approaches based on the multidimensional structure of subjective

well-being originated in the 1980s [29], evolving from a three-dimensional structure to one with four or more components [30].

According to the dominant view, the tripartite vision, adopted in the framework used by the OECD in 2013, assesses subjective well-being in three main dimensions or aspects: (1) Personal satisfaction (Life evaluation), (2) Mood state (affect), and (3) Eudaimonia. These dimensions require specialized exploration and objective well-being such as income level, employment, job stability, and cost of living, among other conditions [31–33].

On the other hand, analyzing the objective well-being factors, there are authors and organizations that explore the theoretical understanding of problems and their practical comprehension for the development of public policies, using measures and modeling behaviors. Examples include [34–39], among others, who even employ structural equation modeling to relate subjective well-being to its direct and indirect drivers such as health, quality of life, social support, income level and other variables. Understanding the evolution of well-being (subjective and objective) during and after the pandemic is highly important, and how individuals have been affected by the rapid changes in economic and social conditions is relevant too. Although there are several studies about the pandemic and its consequences in the world, there is no previous research that analyzes the evolution of the pandemic in two different time lapses considering the economical and psychological contexts using a longitudinal SEM.

### 1.1. Objective Factors: Evolution of Economic and Social Conditions during COVID-19

The first imported case of COVID-19 in Mexico was recorded on 27 February 2020, prompting calls to "follow health protocols". Mexico developed its epidemic control policy during the initial months of the pandemic in the absence of an effective vaccine or treatment for the disease, prioritizing "preventing dispersion through self-care" [39–49].

On 18 March, the first COVID-19-related death occurred in Mexican territory. During the same month, the "National Social Distancing Campaign" was implemented [50], and epidemic control measures were gradually intensified, leading to significant short-term transformations in the economic and social dynamics of Mexicans. The policy pursued by the federal government of Mexico during the initial months of the pandemic was defined in a four-phase strategy, gradually tightening restrictions on people's mobility and economic activity to curb the rate of transmission (See Table 1):

**Table 1.** Phased System of Mexico's Pandemic Control Strategy.

| Phases | 2020 Period | Description |
|---|---|---|
| Import phase | (8 February to 23 March) | The virus enters the national territory, and priority is given to the monitoring and control of dispersed cases associated with the international flow of passengers. |
| Community dispersal phase | (24 March to 20 April) | Actions are taken to control local transmission, suspending educational activities, meetings and crowd events, as well as non-essential economic activities, while preparatory actions are carried out to address the escalation in demand for medical services due to COVID-19 in the main metropolitan areas of the country. |
| Epidemic contagion phase | (21 April to 17 May) | Characterized by the exponential growth of infections, the national healthy distance day is extended, hospital reconversion plans are developed, and more than 65 thousand cases and 9 thousand deaths are reached in the country. |

**Table 1.** *Cont.*

| Phases | 2020 Period | Description |
|---|---|---|
| Epidemiological traffic light phase | (18 May onwards) | A mobility control system and differentiated restrictions are generated between the states of the republic depending on a weighted control of epidemiological indicators that prioritize the balance between mitigating the infection rate and the hospital capacities of the entities to care for the cases, in order to reduce mobility controls regionally and release accumulated economic pressures. |

Source: Self-research.

As seen in Table 1, Mexico's National Health System faced a deficient situation compared to international standards for pandemic response [51]. Mexico allocated less than 6.2% of its GDP to the healthcare sector, nearly 3 percentage points below the average of countries belonging to the Organization for Economic Cooperation and Development [52], which allocated around 8.9%. Additionally, per capita spending was three times lower compared to other OECD countries, accompanied by particularly low public spending, a shortage of specialized healthcare personnel, significant deficiencies in healthcare infrastructure and hospital capacity, and an inadequate healthcare financing system with a high reliance on public assistance over the past decades.

The Mexican Institute for Competitiveness declared by the end of 2018 that Mexico was "on the brink of a public health crisis" [53,54]. This was a result of increasingly significant budget cuts in real terms to the healthcare sector, which amounted to nearly 20% of the annual budget. This made the prospect of a pandemic outbreak in national territory exceptionally complex due to the sudden surge in demand for medical assistance during a pandemic. This underscored the importance of epidemic control based on physical distancing and strategic mobility control.

Furthermore, the national economic structure, income, expenditure, and occupation conditions revealed by the National Occupation and Employment Survey [55] and the National Survey of Income and Expenditures of Households [56] have shown a trend over the past 20 years towards significant labor and income distribution inequality. A large portion of the Mexican population is below the poverty line and experiences a high degree of informal employment. This makes the population vulnerable to income loss due to lockdown measures, rising price levels, scarcity of basic goods and services, and the impact of increased out-of-pocket healthcare expenses.

Given Mexico's vast territorial expanse, economic productive specialization diversity, sociodemographic characteristics, regional dynamics, significant social inequality, and a deficient healthcare system, the country prioritized mobility control, physical distancing, and restrictions as the predominant strategies for infection control. Between July and November 2020, several vaccines internationally began phase III clinical trials, gradually alleviating the long-term stress faced by countries unable to cope with the severity of the pandemic. By 31 December 2020, the WHO included the first COVID-19 vaccine in its emergency use list. On 23 December 2020, Mexico received its first batch of COVID-19 vaccines [57], with the first doses administered on 24 December, marking the beginning of the National Vaccination Strategy.

The National Strategy prioritized the immunization of healthcare workers, older adults, and individuals with comorbidities from December 2020 to April 2021. Table 2 provides a detailed overview of the five stages through which this strategy was implemented in Mexico, starting with those at the highest risk of infection and death and gradually immunizing the population with lower risk of severe COVID-19 consequences by 2022.

**Table 2.** Stages of the National COVID-19 Vaccination Strategy.

| Stage | Period | Population to Immunize |
|:---:|:---:|:---:|
| 1 | December 2020–February 2021 | Frontline health personnel controlling COVID-19 |
| 2 | February–April 2021 | Remaining health personnel and people aged 60 and over |
| 3 | April–May 2021 | People from 50 to 59 years old |
| 4 | May to June 2021 | People from 40 to 49 years old |
| 5 | June 21 to March 2022 | Rest of the population |

Source: Self-research.

With the implementation of the vaccination program, distancing policies were gradually relaxed, creating a context of calm and containment. Although infections persisted and reactivated in waves, the mortality rate and severe illness, as well as hospital occupancy, decreased significantly. This allowed federal entities to normalize their economic and social dynamics.

Regarding the assessment of public health impacts, reports from Johns Hopkins University [58] and the Government of Mexico, as reported through the National Commission for Science and Technology [59,60], revealed that as of 31 October 2020, there were 45,428,731 confirmed cumulative cases and 1,185,721 COVID-19 deaths worldwide, with a global case fatality rate of 2.6%. In Mexico, however, there were 924,962 total cases, 91,753 deaths, and a case fatality rate of 9.8% (See Table 3). This indicated a higher lethality of the pandemic in Mexico, attributed to its poorer living conditions, health infrastructure, and overall healthcare, which indirectly could impact society's mental well-being [61].

**Table 3.** General information on COVID-19 in Mexico and worldwide accumulated as of 31 October 2020.

| | Global | The Americas | Mexico |
|:---|:---:|:---:|:---:|
| Confirmed Cases | 45,428,731 | 19,737,794 | 924,962 |
| Deaths | 1,185,721 | 625,973 | 91,753 |
| Fatality Rate | 2.61% | 3.17% | 9.92% |
| Confirmed Cases during the Last 15 Days | 5,866,774 | | |
| Suspects | | | 358,175 |
| Negatives | | | 1,120,362 |
| Confirmed Active Cases | | | 31,477 |
| | | | 3.4% |

Source: Self-research.

The technical reports from the Ministry of Health [62,63] indicated that at the national level, 51% of cases were predominant among men, with a median age of 43 years. Moreover, just 10 federal entities accounted for over 62% of the cases nationwide. This underscores the high concentration of the pandemic in urban areas characterized by significant economic and social inequality, exacerbated by disparities in the preparedness of their state health systems [64]. Notably, Mexico City alone accounted for 18% of the cases (Mexico City, State of Mexico, Nuevo León, Guanajuato, Sonora, Veracruz, Puebla, Tabasco, Jalisco, and Coahuila).

Results from the Survey on the Economic Impact Generated by COVID-19 in Companies [65] conducted in Mexico with 5671 companies in April and August 2020 revealed that 89.9% of the country's businesses faced some form of impact due to the pandemic during the first half of 2020. They experienced reduced income sources, decreased demand due to changes in consumer preferences and general mobility restrictions, as well as various disruptions in value chains due to shortages of inputs and work products both nationally and internationally.

On average, 15% of the country's economic units were forced to implement staff reduction policies to cope with the income decline, while 14% continued to maintain their workforce but with reductions in compensation levels and employee benefits. The first

three months of the pandemic were the most critical, during which 59.6% of companies faced temporary closures or technical layoffs, a situation that began to improve by April, when 23% of companies were affected for these reasons (See Table 4).

**Table 4.** Effects on business and employment due to pandemic, March–August 2020.

| | **ECOVID-IE** | | |
| --- | --- | --- | --- |
| | **April** | **August** | **Average** |
| Effects on income level | 93.2% | 86.6% | 89.9% |
| Staff reduction | 14% | 16% | 15.0% |
| Reduction of salaries and benefits | 17% | 11% | 14.0% |
| Temporary closures and technical stoppages | 59.60% | 23% | 41.3% |

Source: Self-research.

The pandemic increased business mortality, particularly affecting those in the commercial sector and micro and small enterprises, which represent the largest source of employment and jobs at the national level. In a comparative effort between the periods of January–December 2019 and January–September 2020, using information from the 2019 Economic Censuses and the 2020 Business Demographics Study [66] conducted by the National Institute of Statistics and Geography [67], it can be argued that there was a net decrease in economic units in the country and their ability to provide employment and generate income for Mexicans due to the crisis.

It is estimated that over 1,010,000 establishments permanently closed their doors during the period of January–September 2020, representing 20.8% of the national total. When combined with the birth of new businesses in 2020, this resulted in a net contraction of 8.06% of economic units in Mexico, with micro and small businesses being the most affected. Additionally, the number of employees decreased by 19.68% due to the loss of these vulnerable, family-owned economic units (See Table 5).

**Table 5.** Business mortality and employment contraction due to pandemic, March–September 2020.

| | **Number of Establishments** | **Employed Personnel** |
| --- | --- | --- |
| Initial | 4,857,007 | 14,660,209 |
| Births | 619,443 | 1,231,297 |
| Deaths | 1,010,857 | 2,966,965 |
| Decrease in survivors | | 1,149,494 |
| Current | 4,465,593 | 11,775,047 |
| Percentage change | −8.06% | −19.68% |

Source: Self-research.

The policy of mobility restriction on a global scale and in Mexico reduced the dynamism of economic activity, investment, growth, employment, and jobs. Some sectors, such as the tourism sector, experienced setbacks of more than three decades in just one month of the pandemic, destroying businesses and sources of employment and income for numerous families. Therefore, the fear of infection was socially complemented by the fear of the impossibility of maintaining the economic livelihood of families [68].

In a context of fear and stress, families experienced complex interactions within their households. Non-essential activities with higher added value became remote, while those with lower value simply disappeared, leading to closer and more tense family interactions without a visible way out of the crisis. There were increased reports of experiences of domestic violence.

*1.2. Problem Statement*

In the context of the COVID-19 pandemic, living conditions for the population in Mexico and around the world were drastically affected; social distancing and the risk to

public health emotionally impacted people's lives. In this context, it is necessary to evaluate to what extent this pandemic affected people's emotional health and quality of life.

In the case of Mexico, specifically, the SWLS can capture this effect and allow for the comparison of the emotional state of the population just before the pandemic and at the beginning of it. It is necessary to understand the population's situation and their level of emotional distress in order to develop new concepts and ideas on how to improve their situation in the future. This can lead to the development of public health measures and social coexistence actions to restore society's situation in the new normal imposed by a lingering pandemic.

For this reason, it is necessary to conduct research and analysis of the changing subjective well-being in this context. In this research, we contribute to the discussion by analyzing Mexicans' self-perception of their overall life satisfaction through the data contained in the SWLS. We do this by structurally modeling the data across multiple subscales to explain overall life satisfaction (at two time points) while considering a period before and at the onset of the COVID-19 pandemic.

It is important to note that due to the contagion and impact on the population, as well as the lockdown during this period, we only used the SWLS Scale data for two time points: (1) October 2020 and January 2021. This was justified because these are the only two time points that allow us to contrast the perception of overall life satisfaction during the pandemic and before the pandemic. This study aims, on one hand, to evaluate current overall life satisfaction (during the pandemic) and contrast it with the perception of overall life satisfaction one year earlier (before the pandemic). On the other hand, it considers the validity and reliability of the SWLS, even in the context of a pandemic.

From the problem statement, the following research question emerges: does the SWLS Scale (positive emotional states, negative emotional states, personal satisfaction, eudaimonia, personal well-being and satisfaction with the environment) have a significant influence on the perception of overall life satisfaction both after and during the previous year of the COVID-19 pandemic in Mexico?

### 1.3. Research Questions

Do the subscales of subjective well-being in SWLS have a significant influence on overall life satisfaction in both time periods, contrasting information before the COVID-19 pandemic and its onset, through structural modeling with latent variables?

Is there a significant influence between the factors of subjective well-being (positive emotional states, negative emotional states, personal satisfaction, personal well-being, and satisfaction with the environment) on current overall life satisfaction (during the pandemic) and overall life satisfaction in the previous year, using structural modeling with latent variables?

### 1.4. Objective

Determine the influence of the SWLS Scale on the population's perception of overall life satisfaction both, before and after the COVID-19 pandemic, using a structural equation model with latent variables, using the least partial squares method.

### 1.5. Justification

The present study has at least the following contributions:

(1) This study contributes to understanding how psychosocial factors explain influence overall life satisfaction during the COVID-19 pandemic, both after and in the preceding year.
(2) Methodologically, this study makes a significant contribution by the use of robust multivariate methods and structural modeling with latent variables, specifically using the partial least squares method.
(3) This research contributes to the exploration of new lines of research such as investigating psychosocial factors during times of pandemics.

(4) Additionally, this study contributes to the validation of the SWLS in challenging contexts, such as a pandemic.

(5) Furthermore, this research will facilitate the development of proposals aimed at enhancing well-being and quality of life during catastrophes like the COVID-19 pandemic. These proposals will inform public health initiatives and mental health interventions, and guide policymakers in supporting the population during crisis.

*1.6. Research Hypotheses*

**H1:** *"The dimensions of the SWLS Scale—positive emotional states, negative emotional states, personal satisfaction, eudaimonia, personal well-being, and satisfaction with the environment— significantly impact life satisfaction both during the year preceding and following the COVID-19 pandemic in Mexico."*

The SEM with latent variables in this study included 10 specific hypotheses derived from the general research hypothesis. This proposed model is presented in Figure 1:

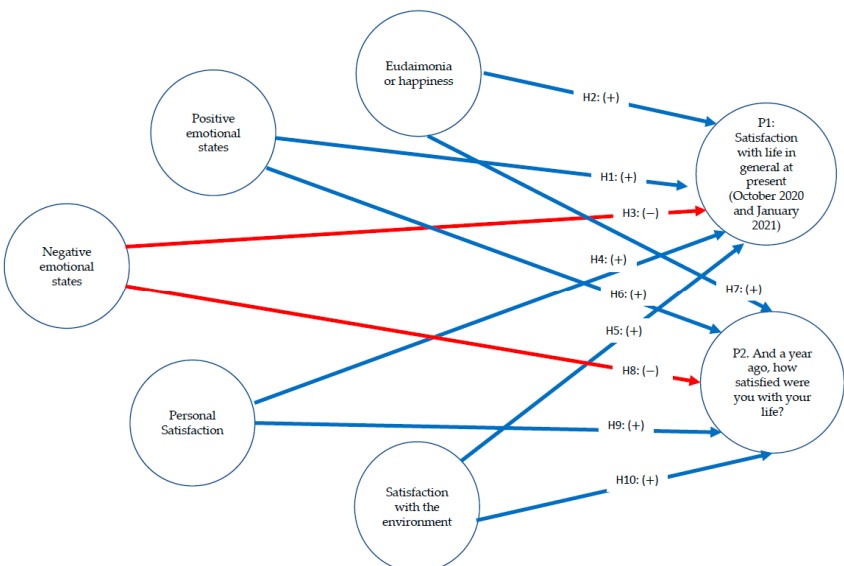

**Figure 1.** Structural equation model with latent variables representing the general research hypothesis. Source: Self-research.

### 1.6.1. Hypotheses Theoretical Background
Subjective Well-Being

Comprehending subjective well-being involves examining individuals' perceptions of their lives, considering a multitude of factors such as emotions, moods, and judgments across diverse domains like family, social interactions, emotions, romantic relationships, personal achievements, mood fluctuations, and self-esteem. This holistic concept encapsulates the overall assessment of one's life and emotional encounters [69].

Subjective well-being is intricately linked to the examination of positive conditions and evaluations within individuals' lives. It focuses on how individuals assess their own well-being and happiness. Often used interchangeably with happiness, subjective well-being seeks to understand and measure the overall satisfaction and contentment individuals experience [70].

Subjective well-being is a multifaceted concept that involves individuals' assessments of various aspects of their lives, including emotional experiences and overall life satisfaction. It is closely linked to happiness and the positive evaluation of one's life conditions. Different authors may emphasize specific aspects or historical origins of this concept, but it generally centers on understanding the positive conditions and evaluations of individuals' lives [71].

Personal Satisfaction

Personal satisfaction is defined as a person's own judgment of life as a whole, which is perceived as appropriate for oneself [72]. This judgment involves comparing one's life as a whole with a personal standard or benchmark. In other words, personal satisfaction is a relative term that considers an individual's unique context, values, and ideas about what constitutes a fulfilling and satisfying life. It emphasizes that what one person finds satisfying and fulfilling may differ from what another person considers satisfying, making it a subjective and context-dependent assessment.

Although the provided excerpt is based on their work, authors emphasize the importance of personal satisfaction as a component of subjective well-being. They suggest that personal satisfaction involves an individual's judgment about their life as a whole, considering their own standards and values. It is a key element in the broader framework of subjective well-being, which encompasses both cognitive and emotional evaluations of one's life [73].

Personal satisfaction refers to an individual's subjective assessment of their life as a whole, with this assessment being relative to their own context, standards, and ideas. It is a fundamental component of subjective well-being and reflects the idea that people's evaluations of their lives can vary significantly based on their personal values and expectations.

Mood State (Affectivity)

Research further conceptualizes mood state, finding that it consists of two opposing dimensions: positive mood state, related to happiness, joy, and delight, and negative mood state, related to sadness, anger, fear, or anxiety. Building on this central idea, the OECD (2013) defines mood state (affectivity) as an emotional state established by the individual at a particular moment, which simultaneously integrates positive and negative elements [74].

Eudaimonia

Eudaimonia is described by [75] as a functional element related to autonomy, competence, a desire to learn, goal orientation, a sense of purpose, resilience, social integration, and altruism. Ref. [76] associates it with an individual's ability to engage in activities aligned with their values. The OECD adopts this dimension as flourishing or the development of an individual's potential.

1.6.2. Previous Research

"Happiness Is the Frequency, Not the Intensity, of Positive versus Negative Affect"

This seminal study examined the components of subjective well-being, specifically the role of positive and negative affect in determining overall happiness. This model suggests that individuals can experience high levels of happiness by frequently experiencing positive affect and minimizing negative affect. One of the central findings of this study is that happiness is more closely related to the frequency of positive affect (i.e., positive emotions) and the absence of negative affect (i.e., negative emotions) rather than the intensity of these emotional experiences. In other words, it is not about how intensely you experience positive emotions, but how often you experience them relative to negative emotions. Also, findings have practical implications for interventions aimed at enhancing well-being. Instead of focusing on intensifying the experience of positive emotions, interventions can be more effective by helping individuals increase the frequency of positive emotions and reduce negative emotions in their daily lives. The study also suggests methodological implications for the assessment of subjective well-being. Researchers and psychologists should consider both the frequency and intensity of positive and negative affect when studying well-being to obtain a more accurate and comprehensive understanding of an individual's happiness [77].

"Review of the Satisfaction with Life Scale"

This paper presents the development and validation of the Satisfaction with Life Scale (SWLS), a widely used instrument for assessing life satisfaction based on Diener's model

of subjective well-being. The scale exhibits good internal consistency, indicating high correlations among its items, thus affirming its reliability. Furthermore, it demonstrates robust convergent validity by positively correlating with other well-being measures and negatively correlating with indicators of distress and negative emotions. Notably, the SWLS displays a one-factor structure, indicating that all its five items contribute to a single overarching construct, namely life satisfaction. This finding reinforces the notion that life satisfaction can be conceptualized as a unidimensional construct in accordance with Diener's model [78].

"Toward a Consensual Structure of Mood"

This study is highly relevant to the tripartite model, exploring the structure of mood encompassing positive and negative affect, which contributes to understanding affective components of well-being. The findings offer insights into the structure of mood and emotions, clarifying their role in subjective well-being. Subjective well-being, according to the tripartite model, comprises three components: life satisfaction, positive affect, and negative affect. The study's results, particularly the distinction between positive and negative affect as two separate dimensions, provide a nuanced understanding of the affective component of well-being. Positive and negative affect represent distinct aspects of emotional experience rather than mere opposites. This study has significantly influenced the development of psychological assessment tools related to mood and emotion, impacting research and clinical practices by emphasizing the importance of separately measuring positive and negative affect for a more precise evaluation of emotional states [79].

"Measuring Positive Emotions"

This study focuses on measuring positive emotions, a central component of Diener's tripartite model. It discusses the development and application of the Positive and Negative Affect Schedule (PANAS) in studying well-being, shedding light on the measurement of positive emotions. The PANAS, with its identification of distinct dimensions of positive and negative affect, has greatly influenced well-being research and provides a valuable tool for assessing individuals' emotional experiences and their relationship to overall life satisfaction [80].

"The Independence of Positive and Negative Affect"

In this research, Diener and Emmons investigate the independence of positive and negative affect, fundamental to the tripartite model, providing evidence for their distinctiveness. The study demonstrates that positive and negative affect are separate emotional states, challenging the traditional idea of them being polar opposites. It suggests that individuals can experience both simultaneously, emphasizing the complexity of human emotions. This research has significant implications for understanding subjective well-being, highlighting that enhancing positive affect does not necessarily diminish negative affect; they can coexist. The main finding suggests that people have a degree of control over their emotional states, able to employ emotion regulation strategies to enhance positive affect and mitigate negative affect even in challenging circumstances [81].

"The Satisfaction with Life Scale and the Emerging Construct of Life Satisfaction"

This paper discusses the evolving concept of life satisfaction within Diener's model and presents findings from studies using the SWLS. It discusses how the understanding of life satisfaction has developed over time and the role of the SWLS in measuring this construct. It provides evidence supporting the validity of the Satisfaction with Life Scale as a reliable measure of life satisfaction. It suggests that this scale has been widely accepted and used in research to assess an individual's overall life satisfaction. It highlights that individual with higher scores on the SWLS tend to have better psychological well-being, including higher positive affect, greater life engagement, and lower negative affect. In summary, this paper underscores the growing recognition of life satisfaction as a central

component of subjective well-being within Diener's model. It validates the Satisfaction with Life Scale (SWLS) as a reliable tool for measuring life satisfaction and highlights its role in assessing and predicting overall well-being. The findings emphasize the utility of the SWLS in cross-cultural research and its significance in understanding individuals' quality of life and happiness [82].

"Personality, Culture, and Subjective Well-Being: Emotional and Cognitive Evaluations of Life"

This study investigates the role of personality and cultural factors in subjective well-being, considering both affective and cognitive evaluations. The study emphasizes that personality traits play a significant role in shaping individuals' subjective well-being. Specifically, it examines the Big Five personality traits (Openness, Conscientiousness, Extraversion, Agreeableness, and Neuroticism) and their relationships with well-being. Findings indicate that personality traits are correlated with life satisfaction and emotional well-being. The research explores the impact of culture on subjective well-being. It finds that cultural factors, such as individualism and collectivism, influence the way individuals evaluate and experience well-being. Cultural norms and values shape emotional and cognitive aspects of well-being. Finally, this study sheds light on the complex interplay of personality and culture in shaping subjective well-being. It emphasizes that subjective well-being includes both emotional and cognitive components, and these components are influenced by personality traits and cultural factors. The findings highlight the need to consider these factors when studying and promoting well-being in diverse cultural contexts [83].

These are some of the studies related to Ed Diener's tripartite model of subjective well-being. Researchers have applied and extended this model in various ways to explore well-being in different populations, cultural contexts, and life circumstances. The model's flexibility and widespread use have contributed to our understanding of the complex nature of human well-being; however, in Mexico, research studies conducted using an SEM remain scarce.

## 2. Materials and Methods

Recent research in social sciences has shifted towards employing more robust probabilistic and mathematical methods. These approaches integrate second-generation multivariate models, enabling the quantitative and explanatory analysis of study phenomena. Specifically, structural equation models (SEMs) are utilized, allowing for the examination of how various exogenous independent variables can elucidate the variance of one or more endogenous or dependent variables. Through the development of simultaneous and joint equations, these structural models provide a detailed understanding of each variable and its causal influence on others [84].

### 2.1. Sample and Data Collection

Data were extracted from the National Self-Reported Well-being Survey (SWLS) in October 2020 and January 2021. This project was approved by the Ethics Committee of the Faculty of Administrative and Social Sciences (NOM-035-STPS-2018-CA-207).

The participants in the study comprised 3615 residents of urban areas in Mexico aged 18 or older, who completed the National Self-Reported Well-Being Survey (SWLS) and the Socioeconomic Questionnaire during the National Consumer Confidence Survey (ENCO) conducted from 4 to 20 January 2021 ($n$ = 1911) and from 1 to 19 October 2020 ($n$ = 1704). Below are the sociodemographic and socioeconomic data of the 3615 participants:

Among the interviewees, there was a higher participation of women (55.4%). The survey collected information from individuals aged 18 to 94, with the most representative age group being the economically active population, from 18 to 65 years old, accounting for 86.9% of the total number of interviewees. Regarding literacy and school attendance, the

sample consisted of 96.7% of people who could read and write messages and 93.8% who no longer attended school.

Concerning their economic activity status, the majority of people stated that they had worked for at least one hour to earn income in the week preceding the survey, representing 58.7% of the total. Meanwhile, 23.4% of interviewees reported being primarily engaged in household chores, and 7.2% were retired or pensioned.

Regarding the occupations of the interviewees, 39.3% chose not to disclose the nature of their occupation. Nevertheless, 29% stated that they were employed in the service sector, 14.5% in the commercial sector as merchants, vendors, or similar roles, and 8.7% as industrial workers. Out of the total, 40.4% reported having salaried employment, either on a wage or salary basis, while 16.2% said they were self-employed.

*2.2. Measurements*

As part of its "measurement of the progress of societies" initiative, the OECD includes the assessment of subjective well-being. Being an OECD member, Mexico participates in this endeavor, with INEGI conducting the SWLS perception survey since 2013 to contribute to this measurement. This survey aligns with similar surveys conducted by other OECD members.

The SWLS, or Self-Reported Well-Being Survey, is integrated as a module of the National Consumer Confidence Survey (ENCO in spanish). These nationwide surveys are conducted quarterly, in January, April, July, and October. However, in 2020, due to the pandemic, the survey wasn't conducted in April and July of that year.

Initially, data from the October 2020 and January 2021 surveys were downloaded, distributed across three tables: housing information, individual household members' details, and responses to the SWLS questionnaire.

While the analysis primarily focuses on data from the third table containing SWLS responses, linking all three tables proves beneficial for obtaining socio-economic, demographic, and territorial information about the respondents.

The linkage between the first two tables relies on six key fields, while the connection between the second and third tables involves seven key fields. Microsoft Access facilitated this linking process, resulting in a unified table with XLS and CSV extensions. These files were subsequently exported to IBM SPSS version 25 and Smart PLS version 3 for structural equation modeling. The data collection comprised two instruments: the National Consumer Confidence Survey (ENCO) and the Self-Reported Well-Being (SWLS) survey. ENCO collects sociodemographic information (age, gender, marital status, income, education level, job characterization), as well as information about housing (location, number of rooms, occupants, type of construction, etc.). These information surveys are conducted four times a year, in January, April, July, and October.

The interviewer collects ENCO survey data and selects a household member aged 18 or older to respond to the SWLS. The interviewer explains to the respondent the purpose of the questionnaire, which is "to collect information on how people feel and their emotional state to generate statistics in comparison with other countries worldwide".

The SWLS Scale consists of two independent questions and three sections of items measuring Eudaimonia (11 items), Affective Balance (10 items), and Satisfaction (12 items).

The independent questions are as follows:

1. How satisfied are you currently with your life?
2. How satisfied were you with your life one year ago?

The responses to these two questions use a Likert-type scale where 0 means completely dissatisfied, and 10 means completely satisfied.

The first section, Eudaimonia, consists of a group of 11 items and refers to "traits of health and emotional strength of the individual". It is answered on a Likert-type scale where 0 means completely disagree, and 10 means completely agree. For example: How much do you agree or disagree with the statement "In general, I feel good about myself"?

The second section, Affective Balance, consists of a group of 10 items and refers to "the emotions people experienced the day before and how much time during the day those emotions lasted". It is also answered on a Likert-type scale where 0 means not at all during the day, 5 means half of the day, and 10 means the entire day. For example: How much of yesterday did you feel in a good mood?

The third section, Satisfaction, consists of a group of 12 items and refers to "the satisfaction that people have with certain domains of satisfaction". It is answered on a Likert-type scale where 0 means completely dissatisfied, and 10 means completely satisfied. It includes twelve items, for example: How satisfied are you with your standard of living?

### 2.3. Data Analysis

The research hypotheses were tested using multivariate statistics, employing structural equation modeling with latent variables, specifically the Partial Least Squares (PLS) method, which allowed for both graphical and statistical analyses of the causal influence of the proposed model.

To assess the instrument's validity, Confirmatory Factor Analysis, Average Variance Extracted (AVE) analysis, and Discriminant Validity analysis were employed. For instrument reliability analysis, Cronbach's Alpha coefficients, Rho A, and Composite Reliability (CR) were used. The structural modeling was developed based on theoretical foundations and the reflective method.

### 3. Results

In this study, we examine the three fundamental dimensions of the SWLS Questionnaire: Affective balance, eudaimonia, and personal satisfaction, forming the basis of our analytical model. We acknowledge the diverse dimensions of positive and negative emotional aspects that influence personal well-being within our analysis. Unlike a confirmatory factor model, our approach does not aim to validate pre-existing constructs. Instead, we opt for an exploratory factor analysis, as detailed in the following sections.

### 3.1. Exploratory Factor Analysis

An initial exploratory factor analysis was conducted using the principal components method, employing the Bartlett's sphericity test and the Kaiser–Meyer–Olkin (KMO) index to assess the sampling adequacy. The KMO index was found to be 0.936, indicating that factor analysis was appropriate. The exploratory factor analysis initially yielded six dimensions with eigenvalues greater than 1, as presented in Table 6:

**Table 6.** Eigenvalues.

| Component | Initial Eigenvalues | | | Sums of Charges Squared of the Extraction | | | Sums of Charges Squared of Rotation | | |
|---|---|---|---|---|---|---|---|---|---|
| | Total | % Variance | % Accumulated | Total | % Variance | % Accumulated | Total | % Variance | % Accumulated |
| 1 | 12.192 | 34.834 | 34.834 | 12.192 | 34.834 | 34.834 | 6.559 | 18.739 | 18.739 |
| 2 | 3.400 | 9.714 | 44.548 | 3.400 | 9.714 | 44.548 | 5.023 | 14.352 | 33.091 |
| 3 | 2.009 | 5.740 | 50.288 | 2.009 | 5.740 | 50.288 | 3.792 | 10.835 | 43.925 |
| 4 | 1.202 | 3.436 | 53.724 | 1.202 | 3.436 | 53.724 | 2.346 | 6.702 | 50.628 |
| 5 | 1.142 | 3.261 | 56.985 | 1.142 | 3.261 | 56.985 | 2.064 | 5.898 | 56.526 |
| 6 | 1.049 | 2.998 | 59.983 | 1.049 | 2.998 | 59.983 | 1.21 | 3.458 | 59.983 |

Source: Self-research.

In Table 6, the generation of six dimensions with their respective variances explained exceeding 30% is observed, reaching a cumulative 59.983%.

Factor analysis was then performed with a varimax rotation to identify the items with their respective dimensions; the result is shown in Table 7 below.

**Table 7.** Rotated component matrix.

| Reagents | Component | | | | | |
|---|---|---|---|---|---|---|
| | **1** | **2** | **3** | **4** | **5** | **6** |
| Positive and negative emotional states<br>P4_7. . . .worried, anxious or stressed? | −0.826 | −0.107 | −0.085 | −0.099 | −0.114 | −0.076 |
| P4_2. . . . calm, calm or calm? | 0.822 | 0.134 | 0.114 | 0.108 | 0.109 | 0.129 |
| P4_1. . . . in good mood? | 0.781 | 0.170 | 0.087 | 0.076 | 0.134 | 0.164 |
| P4_4. . . . Concentrated or focused on what he was doing? | 0.762 | 0.178 | 0.212 | 0.070 | 0.039 | −0.141 |
| P4_3. . . .with energy or vitality? | 0.755 | 0.159 | 0.198 | 0.124 | 0.006 | −0.239 |
| P4_6. . . . moodily? | −0.753 | −0.166 | −0.055 | −0.087 | −0.103 | −0.155 |
| P4_8. . . . tired or without vitality? | −0.742 | −0.125 | −0.153 | −0.106 | 0.019 | 0.291 |
| P4_9. . . .Bored or uninterested in what he was doing? | −0.732 | −0.134 | −0.162 | −0.041 | −0.040 | 0.215 |
| P4_10. . . .sad, depressed or despondent? | −0.708 | −0.160 | −0.149 | 0.001 | −0.173 | 0.154 |
| P4_5. . . .excited or happy? | 0.703 | 0.227 | 0.265 | 0.054 | 0.107 | 0.009 |
| Eudaimonia or happiness<br>P3_4. I have strength in the face of adversity | 0.190 | 0.750 | 0.155 | 0.100 | 0.119 | −0.152 |
| P3_5. I generally feel that what I do in my life is worth it. | 0.191 | 0.715 | 0.217 | 0.029 | 0.155 | 0.062 |
| P3_6. I am a lucky person | 0.178 | 0.693 | 0.278 | 0.020 | 0.144 | 0.069 |
| P3_2. I am always optimistic about my future | 0.202 | 0.683 | 0.219 | 0.116 | 0.275 | −0.107 |
| P3_8. I feel like I have a purpose or a mission in life | 0.105 | 0.669 | 0.325 | 0.060 | 0.014 | 0.129 |
| P3_3. I am free to decide my own life | 0.172 | 0.659 | 0.048 | 0.043 | 0.113 | −0.117 |
| P3_7. Whether things go well or badly depends<br>fundamentally on me. | 0.105 | 0.658 | 0.185 | 0.117 | 0.038 | −0.032 |
| P3_10. Most days I feel like I've accomplished something. | 0.211 | 0.534 | 0.416 | 0.072 | 0.105 | 0.221 |
| Personal satisfaction<br>P5_9. How satisfied are you with your home? | 0.086 | 0.202 | 0.653 | 0.241 | 0.091 | 0.044 |
| P5_3. How satisfied are you with your achievements in life? | 0.181 | 0.339 | 0.621 | 0.058 | 0.273 | 0.061 |
| P5_8. How satisfied are you with the main activity you do<br>(working, household chores, studying, caring for or assisting<br>a family member)? | 0.191 | 0.303 | 0.619 | 0.107 | 0.014 | −0.045 |
| P5_1. How satisfied are you with your standard of living? | 0.163 | 0.234 | 0.607 | 0.121 | 0.386 | 0.043 |
| P5_6. How satisfied are you with the time you have to do<br>what you like? | 0.227 | 0.135 | 0.588 | 0.180 | 0.026 | 0.050 |
| P5_4. How satisfied are you with your personal<br>relationships? | 0.270 | 0.315 | 0.511 | 0.040 | 0.219 | −0.007 |
| P5_5. How satisfied are you with your future prospects? | 0.194 | 0.421 | 0.497 | 0.105 | 0.209 | −0.098 |
| P5_2. How satisfied are you with your health? | 0.283 | 0.271 | 0.378 | 0.079 | 0.252 | −0.315 |
| Satisfaction with the environment<br>P5_11. How satisfied are you with your city? | 0.121 | 0.147 | 0.148 | 0.834 | 0.037 | 0.036 |
| P5_12. How satisfied are you with your country? | 0.143 | 0.106 | 0.061 | 0.816 | 0.144 | 0.016 |
| P5_7. How satisfied are you with your public safety? | 0.072 | −0.002 | 0.225 | 0.647 | 0.093 | −0.004 |
| P5_10. How satisfied are you with your neighborhood? | 0.113 | 0.177 | 0.478 | 0.487 | −0.031 | 0.057 |
| 2. And a year ago, how satisfied were you with your life? | 0.097 | 0.182 | 0.173 | 0.089 | 0.761 | 0.023 |
| 1. Could you tell me on a scale of 0 to 10 how satisfied you<br>are currently with your life? | 0.233 | 0.312 | 0.210 | 0.154 | 0.737 | −0.034 |
| P3_1. In general I feel good about myself | 0.226 | 0.500 | 0.215 | 0.110 | 0.514 | −0.082 |

Extraction method: Principal component analysis. Rotation method: Varimax with Kaiser normalization. The rotation converged in seven iterations. Source: Self-research.

### 3.1.1. First Dimension: Emotional States

This dimension encompassed items related to emotional states, incorporating all of the positive and negative items from the instrument into the structural modeling. Subsequently, a decision was made to separate them into two distinct dimensions in the modeling: Positive Emotional States and Negative Emotional States. These items corresponded entirely to the second block of the SWLS questionnaire known as "Affective Balance", which pertains to the "positive and negative emotional states experienced by the interviewee

the day before the interview". This aligns with previous studies [85,86], noting that, after numerous controversies, it has been concluded that Subjective Well-being includes cognitive-evaluative components (the evaluation of how satisfied a person is with their life, i.e., the relationship between desired and achieved goals) and affective-emotional components (how much a person experiences more pleasant than negative emotions in their life), with the emotional dimension consisting of positive and negative affects. Some authors have argued that these two affects should be considered separate and relatively independent structures [87,88], while other studies have found moderate to high negative correlations between them [89,90].

> *"Furthermore, social support networks, social integration, and trust (dimensions of social capital) show statistically significant positive effects on life satisfaction, positive emotions, and affective balance; and significant negative effects on negative emotion. These results emphasize, first, that the relationship between cultural participation and subjective well-being in Mexico must be studied in its separate categories and components. Second, the social capital approach essentially provides critical insights into understanding the phenomenon."* [91]

The aforementioned findings in previous studies confirmed the decision to divide this dimension into two separate and opposing dimensions regarding Emotional States.

### 3.1.2. Second Dimension: Personal Well-Being

This dimension comprised items related to personal well-being, encompassing all 11 items from the instrument related to the concept of "eudaimonia", except for items 1, 9, and 11. Item P3_9, "Religion is important in my life", addresses a highly important yet controversial topic in Mexico, which could warrant independent and rigorous theoretical, philosophical, theological, and empirical examination. Of the respondents, 72.4% answered that religion is very important in their lives, with values ranging from 8 to 10 on the Likert scale. The mean score was 8.09, with a standard deviation of 2.372 and a mode of 10, ranging from 0 to 10.

Item P3_11, which addresses the difficulty of returning to normalcy after feeling down, was excluded from analysis, as the concept of "normalcy" in the context of a pandemic presents a unique challenge. Furthermore, this item received an average score of 4.41, a standard deviation of 3.086, a mode of 0, and a range from 0 to 10. Notably, 16.4% of respondents fell into the "Totally Disagree" category, 14.4% scored 5 (midpoint), and 11% scored 8. The first group seemed to understand the item and felt that adversity did not affect them, while the second group was indecisive or did not grasp the item's meaning. The third group comprised individuals who were affected by adversity and found it difficult to return to normalcy.

This decision is reinforced by the SWLS itself, with the statement: "Statement 11 refers to something negative, where a higher level of agreement on the scale is worse".

Regarding item P3_1, the exploratory factor analysis grouped it with P1, "Could you rate your current satisfaction with life on a scale from 0 to 10?" and P2, "And a year ago, how satisfied were you with your life?" This grouping is coherent because these items assess satisfaction with oneself and with life. In this study, items P1 and P2 are considered separately as dependent or endogenous variables in the structural modeling with latent variables. Consequently, item P3_1 was not analyzed as part of this dimension. This dimension was labeled "Personal Well-being".

This is consistent with previous research regarding "eudaimonia". The distinction between Subjective Well-being and Psychological well-being, with the former referred to as hedonic and the latter as eudemonic, is a common one. Psychological well-being includes the sense of purpose in life, personal growth, and positive relationships, and explicitly excludes the affective component. However, recent research suggests substantial overlap between the two constructs [92–96].

### 3.1.3. Third Dimension: Personal Satisfaction

This dimension encompasses items related to personal satisfaction, specifically items 7, 10, 11, and 12. These items were defined in the SWLS instrument as "Personal Satisfaction". This alignment is consistent with research. The exploratory factor analysis reclassified these items from their original dimensions into this new "Personal Satisfaction" dimension.

### 3.1.4. Fourth Dimension: Satisfaction with the Environment

The items 11, 12, 7, and 10 were combined and labeled as "Satisfaction with the Environment". P5_11: How satisfied are you with your city? P5_12: How satisfied are you with your country? P5_7: How satisfied are you with your city's security? And P5_10: How satisfied are you with your neighborhood? This dimension was labeled "Satisfaction with the Environment" because, upon reviewing each item, it was evident that they correspond to "satisfaction with the city, country, city security, and neighborhood", in order of decreasing factorial loadings.

### 3.1.5. Fifth Dimension

This dimension corresponds to items used as dependent variables in the present study, namely: P1: Satisfaction with life in general at present and P2: Satisfaction with life in general last year. It also includes item 1: In general, I feel good about myself. It was decided not to consider this dimension as an analytical construct, so P1 and P2 were used separately. There is consistency among the items in P1.

### 3.1.6. Sixth Dimension

The items integrated into this dimension are 9: Religion is important in my life, and 11: When something makes me feel bad, it is difficult for me to return to normal. P3_1: In general, I feel good about myself. This dimension was not analyzed in the present study due to the sensitivity of the topic. This dimension was not included in subsequent analyses in the Structural Equation Models due to the sensitivity of the topic concerning religion, controversy about "returning to normal", and because item P3_1, theoretically, should belong to the second dimension.

A final SEM with latent variables based on the factorial analysis was designed, as shown in Figure 2.

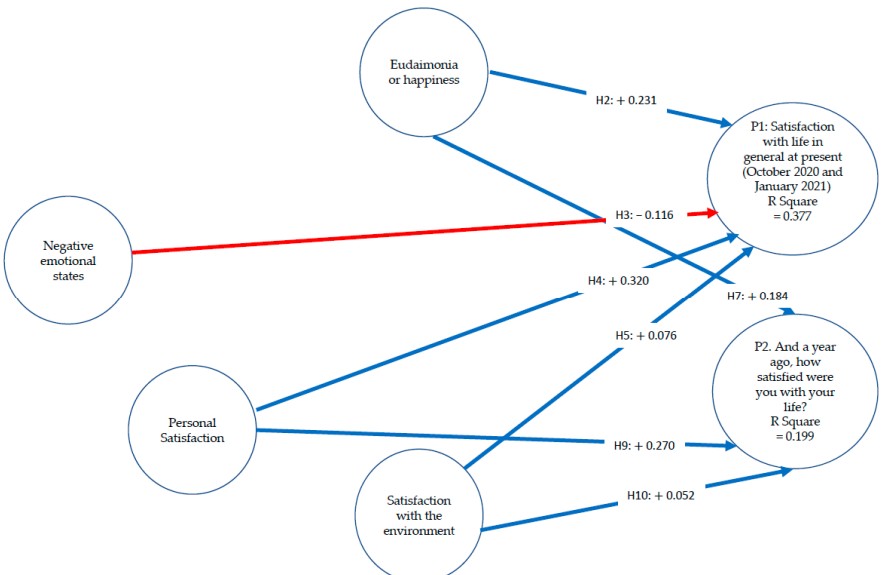

**Figure 2.** Final SEM. Source: Self-research.

The SEM generated 10 specific hypotheses from the research hypothesis, presented in Table 8.

**Table 8.** Specific hypotheses of the SEM.

| Hypotheses | Independent or Exogenous Variables | Description | Expected Influence | Dependent or Endogenous Variables |
|---|---|---|---|---|
| H1: | Positive emotional states | Positive emotional states significantly directly influence how satisfied you are currently with your life (October 2020 and January 2021). | + | P1. How satisfied are you currently with your life? (October 2020 and January 2021) |
| H2: | Eudaimonia or happiness | Eudaimonia or happiness has a significant direct influence on how satisfied you are currently with your life (October 2020 and January 2021). | + | P1. How satisfied are you currently with your life? (October 2020 and January 2021) |
| H3: | Negative emotional states | Negative emotional states significantly inversely influence how satisfied you are currently with your life (October 2020 and January 2021). | - | P1. How satisfied are you currently with your life? (October 2020 and January 2021) |
| H4: | Personal satisfaction | Personal satisfaction has a significant direct influence on how satisfied you are currently with your life (October 2020 and January 2021). | + | P1. How satisfied are you currently with your life? (October 2020 and January 2021) |
| H5: | Satisfaction with the environment | Satisfaction with the environment has a significant direct influence on satisfaction with how satisfied you are currently with your life (October 2020 and January 2021). | + | P1. How satisfied are you currently with your life? (October 2020 and January 2021) |
| H6: | Positive emotional states | Positive emotional states had a significant direct influence on how satisfied you were with your life a year ago. | + | P2. And a year ago, how satisfied were you with your life? |
| H7: | Eudaimonia or happiness | Eudaimonia or happiness had a significant direct influence on satisfaction with how satisfied you were with your life a year ago. | + | P2. And a year ago, how satisfied were you with your life? |
| H8: | Negative emotional states | Negative emotional states significantly inversely influenced how satisfied you were with your life a year ago. | - | P2. And a year ago, how satisfied were you with your life? |
| H9: | Personal satisfaction | Personal satisfaction had a significant direct influence on how satisfied you were with your life a year ago. | + | P2. And a year ago, how satisfied were you with your life? |
| H10: | Satisfaction with the environment | Satisfaction with the environment had a significant direct influence on how satisfied you were with your life a year ago. | + | P2. And a year ago, how satisfied were you with your life? |

Source: Self-research.

*3.2. Descriptive Statistics*

Table 9 presents descriptive statistics, reliability indices, validity measures, and Pearson product-moment bivariate correlations among the various subscales.

**Table 9.** Descriptive statistics, convergent validity, and instrument reliability.

| | Subscales | Mean | Standard Deviation | Cronbach's Alpha | Rho_A | Composite Reliability | AVE | 1 | 2 | 3 | 4 | 5 | 6 | 7 |
|---|---|---|---|---|---|---|---|---|---|---|---|---|---|---|
| 1 | Eudaimonia or happiness | 8.88 | 1.09 | 0.88 | 0.89 | 0.90 | 0.58 | 0.76 | | | | | | |
| 2 | Negative emotional states | 1.56 | 1.55 | 0.87 | 0.87 | 0.91 | 0.66 | −0.392 ** | 0.81 | | | | | |
| 3 | Positive emotional states | 7.61 | 1.84 | 0.89 | 0.89 | 0.92 | 0.70 | 0.465 ** | −0.877 ** | 0.84 | | | | |
| 4 | Satisfaction with the environment | 7.04 | 1.66 | 0.76 | 0.76 | 0.85 | 0.58 | 0.323 ** | −0.222 ** | 0.263 ** | 0.76 | | | |
| 5 | P1. How satisfied are you currently with your life? (October 2020 and January 2021) | 8.13 | 1.73 | 1.00 | 1.00 | 1.00 | 1.00 | 0.520 ** | −0.364 ** | 0.391 ** | 0.310 ** | 1.00 | | |
| 6 | P2. And a year ago, how satisfied were you with your life? | 8.32 | 1.63 | 1.00 | 1.00 | 1.00 | 1.00 | 0.359 ** | −0.235 ** | 0.255 ** | 0.223 ** | 0.588 ** | 1.00 | |
| 7 | Personal satisfaction | 8.44 | 1.18 | 0.83 | 0.84 | 0.88 | 0.55 | 0.698 ** | −0.470** | 0.533 ** | 0.418 ** | 0.564 ** | 0.394 ** | 0.74 |

Source: Self-research. ** The correlation is significant at the 0.01 level (two-sided). AVE Average variance extracted.

### 3.2.1. Mean Scores

The highest mean score was obtained by Personal Well-being ($\bar{x}$ = 8.88, sd = 1.09), followed by Personal Satisfaction ($\bar{x}$ = 8.44, sd = 1.18), with third place occupied by General Life Satisfaction in October 2019 and January 2020 ($\bar{x}$ = 8.32, sd = 1.63), fourth by Current General Life Satisfaction in October 2020 and January 2021 ($\bar{x}$ = 8.13, sd = 1.73), fifth by Positive Emotional States ($\bar{x}$ = 7.61, sd = 1.84), sixth by Satisfaction with the Environment ($\bar{x}$ = 7.04, sd = 1.66), and seventh by Negative Emotional States ($\bar{x}$ = 1.56, sd = 1.55).

### 3.2.2. Reliability

Cronbach's Alpha for all subscales exceeded $\alpha$ = 0.70. Positive Emotional States had the highest value ($\alpha$ = 0.89), followed by Personal Well-being ($\alpha$ = 0.88), Negative Emotional States ($\alpha$ = 0.87), Personal Satisfaction ($\alpha$ = 0.83), and Satisfaction with the Environment ($\alpha$ = 0.76). As for the Rho_a coefficient, Positive Emotional States (Rho = 0.89) and Personal Well-being (Rho = 0.89) tied for first place, followed by Negative Emotional States (Rho = 0.87), Personal Satisfaction (Rho = 0.83), and Satisfaction with the Environment (Rho = 0.76). Composite Reliability scores ranked Positive Emotional States (CR = 0.92) first, followed by Negative Emotional States (CR = 0.91), Personal Well-being (CR = 0.90), Personal Satisfaction (CR = 0.88), and Satisfaction with the Environment (CR = 0.85). These results confirm that the SWLS subscales demonstrate high levels of reliability with strong internal consistency.

### 3.2.3. Validity

Regarding convergent validity, the Average Variance Extracted (AVE) exceeded 0.50 in all subscales. Positive Emotional States had the highest value (0.70), followed by Negative Emotional States (0.66), Satisfaction with the Environment (0.58), Personal Well-being (0.56), and Personal Satisfaction (0.55). Discriminant validity, demonstrated by the square root of AVE, indicated that the correlations within each subscale (diagonal) were higher than the correlations between different subscales, meeting the criterion with values greater than 0.70 in all subscales. Positive Emotional States (0.84) led, followed by Negative Emotional States (0.81), Satisfaction with the Environment (0.76), Personal Well-being (0.76), and Personal Satisfaction (0.74). These results confirm the optimal validity of the instruments, as all AVE values exceeded 0.50, and the discriminant coefficients surpassed 0.70 in all subscales.

### 3.2.4. Correlations

Pearson product-moment correlation coefficients revealed significant inverse correlations between the self-reported subscales and Negative Emotional States. These correlations were as follows: Personal Well-being (r = −0.887 **), Positive Emotional States (r = −0.222 **), Satisfaction with the Environment (r = −0.364 **), Current General Life Satisfaction in October 2020 and January 2021 (r = −0.364 **), General Life Satisfaction in October 2019 and January 2020 (r = −0.235 **), and Personal Satisfaction (r = −0.470 **). Correlations of Personal Well-being with all subscales were as follows: Negative Emotional States (r = −0.392 **), Positive Emotional States (r = 0.465 **), Satisfaction with the Environment (0.329 **), Current General Life Satisfaction in October 2020 and January 2021 (r = 0.520 **), General Life Satisfaction in October 2019 and January 2020 (r = 0.359 **), and Personal Satisfaction (r = 0.698 **). Correlations of Positive Emotional States with all subscales were as follows: Satisfaction with the Environment (r = 0.263 **), Current General Life Satisfaction (r = 0.391 **), General Life Satisfaction in the past year (r = 0.255 **), and Personal Satisfaction (r = 0.533 **). Satisfaction with the Environment was correlated as follows: Current General Life Satisfaction in October 2020 and January 2021 (r = 0.310 **), General Life Satisfaction in October 2019 and January 2020 (r = 0.223 **), and Personal Satisfaction (r = 0.418 **). Correlations of Current General Life Satisfaction in October 2020 and January 2021 with all subscales were as follows: General Life Satisfaction in October 2019 and January 2020 (r = 0.588 **) and Personal Satisfaction (r = 0.564 **). Finally, the correlation between General Life Satisfaction in October 2019 and January 2020 and

Personal Satisfaction was (r = 0.394). These correlation coefficients confirm the expected theoretical consistency among the instrument's subscales.

As we can notice in Table 9, the statistics show acceptable instrument validity and reliability. In Tables 10 and 11, we show the accepted hypotheses' test results as follows, including bootstrapping.

**Table 10.** Accepted hypotheses' test results.

| Hypotheses | Independent or Exogenous Variables | Description | Expected Influence | Beta Coefficient | Dependent or Endogenous Variables | R Square | Decision |
|---|---|---|---|---|---|---|---|
| H2: | Personal welfare | Personal well-being has a significant direct influence on satisfaction with life in general, current October 2020 and January 2021 | + | 0.231 | Satisfaction with life in general, October 2020 and January 2021 | 0.377 | Accept |
| H3: | Negative emotional states | Negative emotional states have a significant inverse influence on satisfaction with life in general, current October 2020 and January 2021 | − | −0.116 | | | Accept |
| H4: | Personal satisfaction | Personal satisfaction has a significant direct influence on satisfaction with life in general, current October 2020 and January 2021 | + | 0.32 | | | Accept |
| H5: | Satisfaction with the environment | Satisfaction with the environment has a significant direct influence on satisfaction with life in general, current October 2020 and January 2021 | + | 0.076 | | | Accept |
| H7: | Personal welfare | Personal well-being had a significant direct influence on satisfaction with life in general last year, October 2019 and January 2020. | + | 0.184 | Satisfaction with life in general last year October 2019 and January 2020 | 0.199 | Accept |
| H9: | Personal satisfaction | Personal satisfaction had a significant direct influence on satisfaction with life in general last year, October 2019 and January 2020. | + | 0.27 | | | Accept |
| H10: | Satisfaction with the environment | Satisfaction with the environment had a significant direct influence on satisfaction with life in general last year, October 2019 and January 2020. | + | 0.052 | | | Accept |

Source: Self-research.

**Table 11.** Bootstrapping.

| | Hypothesis | Original Sample (O) | Sample Mean (M) | Standard Deviation (STDEV) | t Statistics (∣O/STDEV∣) | *p* Values |
|---|---|---|---|---|---|---|
| H2: | Eudaimonia or happiness -> Satisfaction with life in general today, October 2020 and January 2021 | 0.231 | 0.229 | 0.034 | 6.849 | 0 |
| H7: | Eudaimonia or happiness -> Satisfaction with life in general last year, October 2019 and January 2020 | 0.183 | 0.183 | 0.032 | 5.763 | 0 |
| H3: | Negative emotional states -> Satisfaction with current life in general, October 2020 and January 2021 | −0.12 | −0.122 | 0.04 | 3.002 | 0.003 |
| H8: | Negative emotional states -> Satisfaction with life in general last year, October 2019 and January 2020 | −0.009 | −0.011 | 0.042 | 0.219 | 0.827 |
| H1: | Positive emotional states -> Satisfaction with current life in general, October 2020 and January 2021 | −0.005 | −0.007 | 0.043 | 0.122 | 0.903 |
| H6: | Positive emotional states -> Satisfaction with life in general last year, October 2019 and January 2020 | −0.002 | −0.004 | 0.044 | 0.052 | 0.958 |
| H5: | Satisfaction with the environment -> Satisfaction with life in general today, October 2020 and January 2021 | 0.076 | 0.076 | 0.019 | 3.927 | 0 |
| H9: | Satisfaction with the environment -> Satisfaction with life in general last year, in October 2019 and January 2020 | 0.052 | 0.054 | 0.022 | 2.393 | 0.017 |
| H4: | Personal satisfaction -> Satisfaction with life in general today, October 2020 and January 2021 | 0.32 | 0.323 | 0.034 | 9.446 | 0 |
| H10: | Personal satisfaction -> Satisfaction with life in general last year, October 2019 and January 2020 | 0.268 | 0.268 | 0.033 | 8.05 | 0 |

Source: Self-research.

## 4. Discussion

The results of hypothesis testing allowed the following findings:
Hypotheses 1, 6, and 8 were rejected:

**Hypothesis H1:** *"Positive emotional states have a significant direct influence on overall life satisfaction". Respondents in the October 2020 and January 2021 samples considered that positive emotional states did not significantly influence their overall life satisfaction. This was deduced from the P-value exceeding 0.05 in the structural equation model and subsequent bootstrapping, leading to the removal of this variable from the SEM.*

**Hypothesis H6:** *"Positive emotional states have a significant direct influence on overall life satisfaction in the past year". Similarly, respondents in the October 2020 and January 2021 samples (responding with respect to the previous year) believed that positive emotional states did not significantly influence their overall life satisfaction. This was deduced from the p-value exceeding 0.05 in the structural equation model and subsequent bootstrapping, leading to the removal of this variable from the SEM.*

Considering the findings of hypotheses H1 and H6, respondents discarded or did not consider positive emotional states important both at the onset of the pandemic and before it. This suggests that respondents were in a constant state of non-positivity that did not affect their life satisfaction.

**Hypothesis H8:** *"Negative emotional states had a significant inverse influence on overall life satisfaction in the past year". Respondents in the October 2020 and January 2021 samples (responding with respect to the previous year) believed that negative emotional states did not significantly influence their overall life satisfaction in the past year. This was deduced from the p-value exceeding*

*0.05 in the structural equation model and subsequent bootstrapping, leading to the removal of this variable from the SEM. With this hypothesis, respondents maintained negative emotional states only during the period at the start of and during the pandemic in October 2020 and January 2021. However, these negative emotional states were not relevant before the COVID-19 pandemic, so they were excluded from the SEM. They were relevant during the pandemic, as these negative emotional states allowed for reflection on their importance to overall life satisfaction.*

Hypotheses 2–5, 7, 9, and 10 were accepted:

**Hypothesis H2:** *"Eudaimonia or happiness had a significant direct influence on overall life satisfaction in October 2020 and January 2021". The significant direct influence of personal well-being was confirmed based on a standardized regression coefficient of 0.231. This suggests that respondents placed importance on their personal well-being, which considers factors such as optimism, freedom to make life decisions, resilience in the face of adversity, and having a sense of purpose in life. Even during the pandemic, they could be satisfied with their overall life. This indicates a sense of "resilience" among respondents as they faced unknown and adverse circumstances while maintaining a positive outlook.*

**Hypothesis H3:** *"Negative emotional states had a significant inverse influence on overall life satisfaction in October 2020 and January 2021". The significant inverse influence of negative emotional states was confirmed based on a standardized regression coefficient of −0.116. This reaffirms the negative situation experienced by respondents during the pandemic, where negative emotional states could lead to manifestations of worry, anxiety, stress, fatigue, disinterest, sadness, and irritability, which contrast with life satisfaction.*

**Hypothesis H4:** *"Personal satisfaction had a significant direct influence on overall life satisfaction in October 2020 and January 2021". The significant direct influence of personal satisfaction was confirmed based on a standardized regression coefficient of 0.32. This suggests that respondents placed importance on their personal well-being, which takes into account their health and overall balance in economic, social, individual, and psychological aspects during the pandemic, allowing them to be satisfied with their overall life.*

**Hypothesis H5:** *"Satisfaction with the environment had a significant direct influence on overall life satisfaction in October 2020 and January 2021". The significant direct influence of satisfaction with the environment was confirmed based on a standardized regression coefficient of 0.076.*

**Hypothesis H7:** *"Eudaimonia or happiness had a significant direct influence on overall life satisfaction in the past year in October 2019 and January 2020". The significant direct influence of personal well-being was confirmed based on a standardized regression coefficient of 0.0184.*

**Hypothesis H9:** *"Personal satisfaction had a significant direct influence on overall life satisfaction in the past year in October 2019 and January 2020". The significant direct influence of personal satisfaction was confirmed based on a standardized regression coefficient of 0.027.*

**Hypothesis H10:** *"Satisfaction with the environment had a significant direct influence on overall life satisfaction in the past year in October 2019 and January 2020". The significant direct influence of satisfaction with the environment was confirmed based on a standardized regression coefficient of 0.052.*

When the Structural Equation Model analyzed the combined influence of hypotheses, the following was observed:

Hypotheses 2–5 each examined the possibility of the influence of exogenous or independent variables such as personal well-being (0.231), negative emotional states (−0.116), personal satisfaction (0.32), and satisfaction with the environment (0.076) in explaining

overall life satisfaction in October 2020 and January 2021 (during the pandemic). These variables collectively explained approximately 38% of this dependent variable's variance.

On the other hand, Hypotheses 7, 9, and 10 each examined the possibility of the influence of exogenous or independent variables such as personal well-being (0.184), personal satisfaction (0.27), and satisfaction with the environment (0.052) in explaining overall life satisfaction in October 2019 and January 2020 (the year before). These variables collectively explained approximately 20% of this dependent variable's variance.

For a more detailed understanding of the above, it is advisable to review the Table 11.

Eudaimonia or happiness had a greater influence on life satisfaction during the pandemic (0.231) than on life satisfaction a year before (0.184). This could be understood because during the pandemic, people reflected on their emotional health and emotional resilience.

Personal satisfaction also had a greater influence on life satisfaction during the pandemic (0.32) compared to life satisfaction before the pandemic (0.27). This can be explained by the increased personal relationships, personal activities, leisure time, savings, and family economy, as well as the lack of commuting between home and work or school during the pandemic, among other factors.

Negative emotional states only significantly affected current life satisfaction (−0.116), potentially reducing it. This could be understood as "higher scores of negative emotional states leading to lower life satisfaction during the pandemic". An important point is that this variable did not influence life satisfaction before the pandemic (the hypothesis was rejected in the SEM). On the other hand, positive emotional states did not influence life satisfaction either before the pandemic or during the pandemic (these hypotheses were eliminated in the SEM). This can be interpreted as people, in a disastrous situation of risk, death, disillusionment, hopelessness, uncertainty, and increased discomfort, eliminated positive emotional states from their subjective perception, considering them ephemeral, non-permanent, and irrelevant in life.

Satisfaction with the environment had little influence, almost to the same extent, both on life satisfaction during the pandemic (0.076) and in the situation before the pandemic (0.052). The questionnaire's questions for the interviewees referred to satisfaction with their city, their country, their neighborhood, and public safety.

The exploratory factor analysis grouped items P3_9 and P3_11 into a sixth dimension, with a loading of less than 3 in the percentage of explained variance. Therefore, in the present research, it was decided not to consider these items, as listed below:

- Item P3_9: "Religion is important in my life".
- Item P3_11: "When something makes me feel bad, it is hard for me to return to normal".

As for items P5_7, P5_10, P5_11, and P5_12, the exploratory factor analysis grouped them into a fourth dimension with a loading of 3.436 of the percentage of explained variance. These items should belong to the personal satisfaction factor in the SWLS. However, in the present research, we placed them in a factor called 'satisfaction with the environment'. These items are as follows:

- P5_7: "How satisfied are you with public safety?"
- P5_10: "How satisfied are you with your neighborhood?"
- P5_11: "How satisfied are you with your city?"
- P5_12: "How satisfied are you with your country?"

## 5. Conclusions

This study successfully determined the significant influence of SWLS subscales on overall life satisfaction at two time points (October 2020 and January 2021) during the COVID-19 pandemic in Mexico. The study involved items referring to the most complex and devastating period in terms of global public health during the COVID pandemic and

the immediately preceding year, when there was no social reason to anticipate a pandemic of such magnitude and the global policies taken to control its effects.

The results allow us to understand that with the onset of the pandemic, the parameters by which Mexicans construct their life satisfaction were modified. Before the pandemic, life satisfaction was primarily explained by personal appreciation, mainly due to positive factors like happiness, personal satisfaction, and satisfaction with the environment. However, with the emergence of the pandemic and its economic, social, and public health impacts, individuals had to restructure their understanding of life. This shift caused negative emotional states to explain 30% of their overall life satisfaction.

Based on data from the SWLS in Mexico, the method used in the study, structural equation modeling, revealed the extent to which Mexicans explained their life satisfaction before and after the pandemic. It demonstrates the magnitude of the change and what can be understood as a widespread emotional impact on society due to the looming health risk caused by the pandemic.

The SEM results support findings from previous research, such as [97–99], who establish links between SWLS dimensions and life satisfaction. Variables like social support, happiness, health, and other factors were associated with personal mood states and individual eudaimonic well-being as well as self-perception of well-being. Moreover, numerous prior studies, previously cited, have underscored the notable and enduring effects of individual constraints resulting from the pandemic on both well-being and mental negative states across various age demographics, with a particular emphasis on younger individuals. Specifically, there have been reports of heightened stress, anxiety, sleep disturbances, feelings of confusion, pessimism, and symptoms of obsessive–compulsive disorders.

The theory was confirmed in the findings, going beyond the general relationships presented by these authors. The study reveals the sensitivity of individuals to changes in their life environment. An adverse, dangerous, and socially distant environment led a predominantly satisfied society to integrate negative emotional states into their subjective well-being, making them highly significant in self-perception.

The SEM explains less than 40% of life satisfaction during the pandemic and less than 20% for life satisfaction before the pandemic. PLS-SEM results show that emotional balance was disrupted, becoming more sensitive to negative emotional states. Even positive emotional states were removed from the model. The absence of a solution to the crisis gradually eroded faith and optimism in a Mexican society that was once satisfied with itself and its well-being. The post-pandemic emotional state is explained more by the complex and abnormal context of the crisis than by the normal emotional trend.

The pandemic spurred individuals, society, businesses, and governments to contemplate the significance of life and death, as well as various aspects such as quality of life, work, relationships, family, time, and health. This collective reflection led to an emotional journey for everyone involved. The psychological impact of subjective factors became more evident when comparing pre-pandemic conditions to those during the pandemic.

Unlike other studies, the SEM composition for Mexico with SWLS, before and after the pandemic, led to the separation of emotional states. These items were formulated years before the pandemic as part of the Personal Satisfaction construct. However, the atypical conditions of the pandemic's evolution in Mexico in October 2020 and January 2021 caused these four items to behave differently. Therefore, they were grouped into another construct. This underscores the need to vary the number of constructs and assessment dimensions according to the context and societal impact on emotional states and overall satisfaction.

Exploratory factor analysis integrated question P3_1, "In general, I feel good about myself", along with questions P1 and P2, which were treated as dependent or endogenous variables in this study. These questions asked about current and past life satisfaction. Due to the study's focus on comparing life satisfaction between the two time points, question P3_1 was removed from the SEM analysis.

It is worth mentioning that for the Mexican context, the exploratory factor analysis included item P3_9, "Religion is important in my life", along with P3_11, "When something

makes me feel bad, it is hard for me to return to normal", as a separate dimension from what was theoretically considered Eudaimonia. This suggests a need to delve deeper into questions about religion's role in people's lives and the evolving concept of "normalcy" in a post-pandemic world.

The model shows that a significant portion of well-being is unexplained by the SWLS questionnaire, depending on other conditions that impact individuals. Leveraging the insights from the pandemic and its economic, social, and public health lessons, the SWLS could be enhanced with additional modules to explain the causes of emotional fluctuations over time, related to environmental changes affecting the surveyed individuals.

Recommendations for questionnaire modification include the need for clear, specific questions to ensure respondents' clarity and avoid confusion, particularly in areas related to public safety and social security.

The study demonstrates that the SWLS, developed before the pandemic, may not have fully captured the changes in people's perceptions and feelings due to the pandemic. Therefore, the questionnaire should be updated to better align with the post-pandemic reality and the experiences of individuals.

The SWLS allows an understanding of subjective well-being from a self-referenced, abstract, and idealized perspective. However, the questionnaire's limited analytical capacity to integrate sufficient features for understanding the causality of these perceptions or providing material context explanations can be improved.

The SWLS is an instrument developed in a pre-pandemic context, representing subjective well-being conditions under normal societal conditions. Nonetheless, evidence of well-being sensitivity to environmental changes, as seen during the pandemic, suggests the need to supplement the questionnaire with items exploring well-being vulnerabilities and transformations in changing contexts.

The questionnaire should also explore responses in rural environments to evaluate how urban and rural living conditions affect self-reported well-being standards, as the current application focuses solely on urban and metropolitan settings.

In conclusion, this study adds a comprehensive perspective to our understanding of the impact of the COVID-19 pandemic on the subjective and objective well-being of the Mexican population. By employing the SWLS questionnaire and utilizing structural equation modeling, the research captures the intricate shifts in individuals' perceptions of life satisfaction during the most challenging phases of the pandemic. Also, as we could see on the contextual background, the impact of objective well-being and its evolution was well documented. In the objective scenario, evolution showed a negative result for most businesses and organizations, individuals and their families.

On the subjective factors, the findings emphasize the dynamic nature of well-being, illustrating a transition from a predominantly positive pre-pandemic scenario to a post-pandemic reality where negative emotional states significantly contribute to overall life satisfaction. This aligns with the broader literature on well-being and reinforces the need for a nuanced approach in evaluating societal satisfaction during times of crisis.

The study's outcomes are consistent with previous research by authors underlining the robust connections between SWLS dimensions and life satisfaction. The identified variables, including social support, happiness, and health, corroborate the importance of these factors in influencing individual perceptions of well-being [100].

Moreover, the SEM results presented in this study contribute to the existing body of knowledge by unveiling the profound emotional impact of the pandemic on Mexican society. The sensitivity of individuals to changes in their environment and the substantial role played by negative emotional states in shaping well-being underscore the need for a more comprehensive understanding of the intricate interplay between external events and subjective satisfaction.

Despite the insightful findings, it is crucial to acknowledge the limitations of the SWLS instrument in capturing the entirety of this transformation. The study echoes

recommendations from other scholars, emphasizing the necessity for questionnaire updates to better align with the post-pandemic reality and evolving experiences of individuals.

In summary, this research reinforces the importance of reevaluating and adapting our instruments for well-being assessment in response to societal transformations. By embracing the lessons learned from the pandemic, we have an opportunity to enhance the SWLS questionnaire and other similar tools, providing a more accurate reflection of individuals' subjective well-being in the ever-evolving post-pandemic landscape.

Conversely, these findings illuminate the adverse effects on life satisfaction in the aftermath of the pandemic and their implications for Mexico. It is imperative to initiate measures aimed at enhancing overall opportunities for businesses and small entrepreneurs, providing them with avenues to recuperate and ameliorate their financial standing.

*5.1. Recommendations*

This pandemic should have important lessons learned by every country. For Mexico, enhancing mental health support should be a paramount concern. Ensuring widespread access to mental health services and bolstering support systems for individuals and communities during times of crisis ought to feature prominently on every government agenda and within educational and health institutions all over the country. Thoughtfully crafted strategies, specifically tailored to address mental health needs, must be meticulously designed and earnestly implemented.

On the other hand, the development of resilience and effective coping mechanisms across all tiers as emergency responses to future crises should be promoted from individual to societal levels. Comprehensive education and training programs covering stress management, mindfulness, and other valuable techniques to empower individuals in navigating adversity effectively should also be offered.

Also, strengthening social support networks and community connections to foster solidarity and mutual aid during crises will be an enormous positive change. Encourage individuals to reach out to friends, family, and neighbors for support, and facilitate the formation of support groups and community organizations. Access to accurate, reliable information from trusted sources is relevant. Invest in public health communication campaigns that provide clear guidance and updates on the evolving situation, while also addressing concerns and dispelling myths [99].

Simultaneously, government should recognize and address the disproportionate impact of crises on vulnerable populations, including low-income communities, marginalized groups, and frontline workers, implementing policies and programs aimed at reducing socioeconomic disparities and ensuring equitable access to resources and support services in a more efficient way.

Likewise, encouraging collaboration and coordination among government agencies, healthcare organizations, community groups, and other stakeholders involved in crisis response should be of great importance. Information sharing, joint planning, and collective action to address challenges and maximize impact should be implemented [100].

In order to prevent distress from containment, previous research recommends including psycho-educational interventions and adopting appraisal strategies that improve perceived self-efficacy.

To summarize, identifying and addressing vulnerabilities in existing systems and structures to build resilience against future crises should be a priority. This may involve investing in disaster preparedness, strengthening supply chains, diversifying economic sectors, and promoting sustainable development practices. Developing a campaign focused on hope and positivity amidst adversity by highlighting stories of resilience, compassion, and solidarity would be helpful. Encourage acts of kindness, volunteerism, and community engagement to inspire optimism and foster a sense of collective purpose.

Finally, by implementing these recommendations, policymakers, healthcare professionals, community leaders, and individuals would greatly help mitigate the impact of future crises on well-being and build more resilient societies.

*5.2. Opportunities for Future Research*

Long-Term Impact Analysis: Future research endeavors could delve into the enduring repercussions of the pandemic on life satisfaction, offering insights into how individuals and communities navigate the evolving landscape over an extended period. Resilience Factors: Exploring potential resilience factors contributing to a rebound in life satisfaction post-pandemic could provide a more comprehensive understanding of coping mechanisms and adaptive strategies. Comparative Analysis: Conducting a comparative analysis between urban and rural populations would offer valuable insights into how diverse socio-environmental contexts may have varying impacts on subjective well-being. Instrument Enhancement: Future research should consider refining well-being assessment instruments to capture emerging dimensions that may have surfaced during the pandemic but are not adequately represented in existing tools. Likewise, future studies should aim for greater diversity in both demographics and geographical representation to ensure the relevance of their findings. This includes not only focusing on urban areas but also considering underdeveloped and isolated regions of the country. Deepening into a longitudinal study, it would be most appropriate to include a study with three time points: before, during, and after COVID-19 in the SEM, reflecting significant differences and seeking their consistency or lack thereof.

On the other hand, enhancing our understanding of long-term effects, identifying resilience factors, conducting comparative analyses, and refining assessment instruments are crucial steps in advancing the field of well-being in future research. Future research also is recommended on the moderating role of self-efficacy on the psychological well-being of populations with isolation restrictions when managing their fears about a crisis. Lastly, future research should delve deeply into the specific policy recommendations or interventions that could be most effective in this context.

For future studies, it could be beneficial to explore a longitudinal design incorporating pre- and post-COVID-19 pandemic phases. This approach would allow for a more in-depth examination of significant differences in both the scores of each SWLS subscale and the SEM itself. The aim would be to assess its explanatory consistency or inconsistency concerning current and overall life satisfaction questions across different time frames, including the previous year. Such a study would involve at least three time points: pre-COVID-19 (2019), during COVID-19 (years 2020 and 2021), and post-COVID-19 (2023).

*5.3. Limitations of the Current Study*

Government-provided data might not have always been accurate or complete due to reporting errors, inconsistencies, or delays in data collection and processing. This could have led to inaccuracies in the analysis and interpretation of findings. Similarly, the government might have had motives to present data in a certain way to portray their handling of the pandemic positively. Consequently, there might have been biases in the reported data, such as underreporting of cases or deaths, which could have affected the validity of the study's conclusions. In addition, there might have been a lag between when data were collected by the government and when they became available for research purposes. This could have impacted the timeliness of the analysis and the ability to detect emerging trends or changes in the pandemic over time.

Furthermore, government data might not have captured regional variations in the spread and impact of COVID-19, particularly in areas with decentralized health systems or uneven data reporting practices across regions. This could have limited the generalizability of the study's findings to different geographic areas. Overall, researchers need to be aware of its limitations and exercise caution in their analysis and interpretation of its findings. This research used a longitudinal design, spanning across two distinct periods during the COVID-19 timeline. However, a limitation of this approach is that these periods occurred at different points in time very close to each other. The intention was to compare the effects of COVID-19 during two separate time periods, during and after the COVID-19 pandemics.

However, to examine changes over time or assess the progression of certain variables, such a short period of time could be a limitation.

It is important to note that in the present study, we did not design the instrument ourselves. Instead, we could only access the INEGI database to analyze the data using a Structural Equation Model. Therefore, we were unable to take any measures to mitigate response bias.

**Author Contributions:** I.A.M.-M., E.M.-T., J.P.R.-D., A.D. and G.E.S.-A. conceived and designed the study. I.A.M.-M., E.M.-T., J.P.R.-D., A.D. and G.E.S.-A. participated in the acquisition of data. I.A.M.-M. analyzed the data. B.R.G.-R. gave advice on methodology. I.A.M.-M. and B.R.G.-R. drafted the manuscript, and I.A.M.-M., E.M.-T., J.P.R.-D., A.D. and G.E.S.-A. revised the manuscript. All authors have read and agreed to the published version of the manuscript.

**Funding:** This research received no external funding.

**Institutional Review Board Statement:** This project was approved by the Ethics Committee of the Faculty of Administrative and Social Sciences according to approval NOM-035-STPS-2018-CA-207. However, it is crucial to note that this project exclusively utilizes publicly available statistical information that neither collects nor records personal identification data. Consequently, the nature of the data employed renders the requirement for an informed consent statement unnecessary.

**Informed Consent Statement:** Not applicable in this instance, as the research exclusively utilized public information that did not involve the collection or recording of personal identification details.

**Data Availability Statement:** The data are available from the corresponding author upon reasonable request.

**Conflicts of Interest:** The authors declare no conflicts of interest.

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
