# Peer review of "Assessing the Impact of COVID-19 on Subjective Well-Being and Quality of Life in Mexico: Insights from Structural Equation Modeling"

_covid, doi:10.3390/covid4050041_

Round 1

Reviewer 1 Report

See my report.

The article is very interesting and give valuable information regarding the influence of Covid 19 on the emotional well-being and overall quality of life in Mexiko.

The study is well executed, designed and reported. The research problem is properly formulated, and the aims are stated clearly. The research design is applicable and the statistical methods are explained well. The conclusion is clear and reflects the aim of the study.

There are however some minor corrections to be done:

1) Refrences: Several references are older than 5 years. Please update where possible.

2) In some of the tables some words has been cut off: for example table 7 see row 4) and table 8 ( seeH5). Please check fond and all tables.

3) From the authors experience on Covid 19 I would like to see some valuable recommendations to which could be helpful in future crises like Covid 19.

Author Response

Dear reviewer,

we appreciate your valuable time, comments and feedback to improve the quality in our manuscript. In next paragraphs we inform you how we modified the paper because we marked in yellow all the changes that were made.

Your comment appears in black words, while our responses are in italic for a better understanding.  

REVIEW 1 POINT-BY-POINT RESPONSE:

  1. References: Several references are older than 5 years. Please update where possible.

Response: In addressing this concern, we meticulously reviewed the references cited in our study. Our analysis revealed that before replacing new references, approximately 34% of the references exceeded the 5-year threshold, while the majority, comprising 65%, were within the range of 2018 to 2024. Recognizing the importance of sourcing information from up-to-date and reliable sources, we endeavored to ensure that our citations were grounded in the most current research available. While maintaining the foundational principles, we strategically replaced references with newer ones where feasible, without compromising the integrity or coherence of the document. The following references were updated: 4,5,6,7,10,24,33,53,71,72 and 77. Furthermore, we needed to keep references related to government databases or public policies that were not possible to be replaced.

In some of the tables some words have been cut off: for example, table 7 see row 4) and table 8 (seeH5). Please check fond and all tables.

Response: To address this issue, we conducted a thorough review of all tables included in the document. Upon identifying instances where words were cut off or formatting issues were present, we took proactive measures to rectify the situation. Specifically, we replaced all tables with editable versions, allowing for adjustments to font size, layout, and other formatting elements as needed. By implementing this solution, we aimed to ensure that all tables are presented in a clear and legible format, thereby enhancing the readability and professionalism of the document. Additionally, this approach facilitates easier editing and updates in the future, should the need arise.  Additionally, we will attach the new version of the tables and figures in Excel and power point.

  1. From the authors experience on Covid 19 I would like to see some valuable recommendations to which could be helpful in future crises like Covid 19. Response:  In response to this suggestion, we have incorporated a dedicated section of recommendations in our study. Beginning at point 5.1, we have provided a comprehensive set of valuable recommendations drawn from the authors' experiences with COVID-19. These recommendations encompass various aspects, including preparedness strategies, response measures, communication protocols, and community resilience-building initiatives. By sharing insights gained from navigating the challenges of COVID-19, we aim to offer practical guidance and actionable recommendations to support future crisis management efforts. This addition strengthens the utility and relevance of our study by providing actionable insights that can inform decision-making and enhance preparedness for similar crises in the future. This section will be found from line 1026 to 1069.

 The point by point reply to all the reviewers is attached bellow. thanks for your time and attention.  we remain pending on your answer.

Reviewer 2 Report

Given the comprehensive nature of the document, here's a precise review focusing on the strengths, weaknesses, limitations, and suggested improvements for the manuscript titled "Shifts in subjective well-being dynamics and quality of life before and After COVID-19: A structural Model Approach" by Ignacio Alejandro Mendoza-Martínez and colleagues.

Strengths:

Comprehensive Analysis: The study provides a detailed investigation of the impact of COVID-19 on subjective well-being and quality of life in Mexico using structural equation modeling, which offers nuanced insights into the complex relationships between various factors affecting well-being during the pandemic.

Robust Methodology: The methodology, particularly the use of structural equation modeling with latent variables and the employment of a large sample size from the National Self-Reported Well-being Survey (SWLS and ENCO), enhances the reliability and validity of the findings.

Significant Findings: The study's findings, revealing a considerable variance in overall life satisfaction attributable to personal well-being, personal satisfaction, satisfaction with the environment, and negative emotional states, contribute valuable insights into the field of health economics and public health.

Reliability and Validity: The instruments used for data collection demonstrated high levels of reliability and validity, with Cronbach's Alpha, Rho_A, Composite Reliability, and Average Variance Extracted (AVE) all indicating strong internal consistency and construct validity.

Weaknesses and Limitations:

Cross-Sectional Design: The cross-sectional nature of the study limits the ability to infer causality and temporal changes in well-being and quality of life.

Geographical and Demographic Scope: The focus on urban areas in Mexico might limit the generalizability of the findings to rural populations or to different cultural contexts.

Potential Response Bias: The reliance on self-reported measures introduces the possibility of response bias, which might affect the accuracy of the findings.

Limited Discussion on Methodological Challenges: The manuscript could benefit from a more detailed discussion on the challenges encountered in employing structural equation modeling in this context and how these were addressed.

Recommendations for Improvement:

Enhancing Generalizability: Future research should aim to include a more diverse demographic and geographical sample to improve the generalizability of the findings.

Longitudinal Design: Implementing a longitudinal study design could provide insights into the temporal dynamics of subjective well-being and the long-term effects of the COVID-19 pandemic.

Addressing Response Bias: The authors should consider methods to minimize potential response bias, such as anonymizing responses or employing triangulation with qualitative data.

Expanding the Discussion on Limitations: The manuscript would benefit from a more comprehensive discussion on the limitations of the study, including the implications of the cross-sectional design and the potential biases inherent in self-reported data.

Conclusion:

The manuscript makes a significant contribution to understanding the impacts of COVID-19 on subjective well-being and quality of life in Mexico. Its methodological rigor and the relevance of its findings position it as a valuable addition to the literature. However, addressing the identified weaknesses and limitations could enhance its robustness and applicability. With these improvements, the study would be a strong candidate for publication, offering important insights for public health policy and future research in the field.Given the comprehensive nature of the document, here's a precise review focusing on the strengths, weaknesses, limitations, and suggested improvements for the manuscript titled "Shifts in subjective well-being dynamics and quality of life before and After COVID-19: A structural Model Approach" by Ignacio Alejandro Mendoza-Martínez and colleagues.

Strengths:

Comprehensive Analysis: The study provides a detailed investigation of the impact of COVID-19 on subjective well-being and quality of life in Mexico using structural equation modeling, which offers nuanced insights into the complex relationships between various factors affecting well-being during the pandemic.

Robust Methodology: The methodology, particularly the use of structural equation modeling with latent variables and the employment of a large sample size from the National Self-Reported Well-being Survey (SWLS and ENCO), enhances the reliability and validity of the findings.

Significant Findings: The study's findings, revealing a considerable variance in overall life satisfaction attributable to personal well-being, personal satisfaction, satisfaction with the environment, and negative emotional states, contribute valuable insights into the field of health economics and public health.

Reliability and Validity: The instruments used for data collection demonstrated high levels of reliability and validity, with Cronbach's Alpha, Rho_A, Composite Reliability, and Average Variance Extracted (AVE) all indicating strong internal consistency and construct validity.

Weaknesses and Limitations:

Cross-Sectional Design: The cross-sectional nature of the study limits the ability to infer causality and temporal changes in well-being and quality of life.

Geographical and Demographic Scope: The focus on urban areas in Mexico might limit the generalizability of the findings to rural populations or to different cultural contexts.

Potential Response Bias: The reliance on self-reported measures introduces the possibility of response bias, which might affect the accuracy of the findings.

Limited Discussion on Methodological Challenges: The manuscript could benefit from a more detailed discussion on the challenges encountered in employing structural equation modeling in this context and how these were addressed.

Recommendations for Improvement:

Enhancing Generalizability: Future research should aim to include a more diverse demographic and geographical sample to improve the generalizability of the findings.

Longitudinal Design: Implementing a longitudinal study design could provide insights into the temporal dynamics of subjective well-being and the long-term effects of the COVID-19 pandemic.

Addressing Response Bias: The authors should consider methods to minimize potential response bias, such as anonymizing responses or employing triangulation with qualitative data.

Expanding the Discussion on Limitations: The manuscript would benefit from a more comprehensive discussion on the limitations of the study, including the implications of the cross-sectional design and the potential biases inherent in self-reported data.

Conclusion:

The manuscript makes a significant contribution to understanding the impacts of COVID-19 on subjective well-being and quality of life in Mexico. Its methodological rigor and the relevance of its findings position it as a valuable addition to the literature. However, addressing the identified weaknesses and limitations could enhance its robustness and applicability. With these improvements, the study would be a strong candidate for publication, offering important insights for public health policy and future research in the field.

I propose to consult and add the following article:

 https://doi.org/10.3390/ijerph20075294

The work cannot be published unless the tables and figures are improved. Please review the changes in the title, introduction and tables before publication. These minor changes should be made

The work cannot be published unless the tables and figures are improved. The work cannot be published unless the tables and figures are improved. Please review the changes in the title, introduction and tables before publication. These minor changes should be made

Author Response

Dear reviewer,

we appreciate your valuable time, comments and feedback to improve the quality in our manuscript. In next paragraphs we inform you how we modified the paper because we marked in yellow all the changes that were made.

Your comment appears in black words, while our responses are in italic for a better understanding.  

REVIEW 2 POINT-BY-POINT RESPONSE:

  1. Enhancing Generalizability: Future research should aim to include a more diverse demographic and geographical sample to improve the generalizability of the findings.

Response: In addressing the issue of enhancing generalizability, we have acknowledged the importance of diversifying the demographic and geographic composition of future research samples. By incorporating this consideration into our discussion on future research areas and limitations, we recognize the significance of ensuring that findings can be applied to a more representative population. This acknowledgment underscores our commitment to advancing research practices that promote inclusivity and broaden the applicability of study outcomes.  This section will be found from lines 1082-1104.

  1. Longitudinal Design: Implementing a longitudinal study design could provide insights into the temporal dynamics of subjective well-being and the long-term effects of the COVID-19 pandemic.

Response:  taking your recommendation into consideration: we added this part in the opportunities for future research section: For future studies, it is recommended to explore a longitudinal study where a pre- and post-COVID-19 pandemic section would be considered. This would reflect greater depth in contrasting significant differences both in the scores of each of the BIARE subscales, as well as in the SEM model itself, seeking its explanatory constancy or non-constancy with respect to the current and general life satisfaction questions and in the previous year. Said study would be carried out taking into account at least three time points, being: before Covid 19 (2019), during Covid 19 (years 2020 and 2021), and after Covid 19 (2023). This will be found from lines 1082-1104.

Addressing Response Bias: The authors should consider methods to minimize potential response bias, such as anonymizing responses or employing triangulation with qualitative data.

Response:  taking your recommendation into consideration, we added this part in the limitations section: It is important to mention that in the present study we did not design the instrument ourselves and we were only able to download the database from INEGI to analyze the data using a Structural Equation Model, so we could not do anything to reduce response bias.  This text will be found from lines 1108-1128.

  1. Expanding the Discussion on Limitations: The manuscript would benefit from a more comprehensive discussion on the limitations of the study, including the implications of the cross-sectional design and the potential biases inherent in self-reported data.

Response:   The discussion of limitations was broadened to encompass the implications of the longitudinal design, which involved comparing two samples from different time periods in 2020 and 2021. However, we also addressed the limitations of the relatively short interval between each sample collection. Additionally, we identified an opportunity for future research to conduct longitudinal analyses by comparing samples from before, during, and after a crisis to provide more robust insights. As mentioned above, limitations section will be found from lines 1108-1128.  We also added this comment from lines 1126-1129: “It is important to note that in the present study, we did not design the instrument ourselves. Instead, we could only access the INEGI database to analyze the data using a Structural Equation Model. Therefore, we were unable to take any measures to mitigate response bias”

  1. I propose to consult and add the following article:

 https://doi.org/10.3390/ijerph20075294

Response:    In response to the suggestion to incorporate the referenced article into our study, we have expanded the discussion of future research and recommendations to encompass the implications of the proposed theory of Cognitive Appraisals on perceived self-efficacy and distress during the COVID-19 pandemic. Drawing insights from the suggested article, we have considered the negative psychological states that affect populations during crisis isolations, such as stress, anxiety, and uncertainty. Moreover, we have incorporated recommendations outlined in the article into our own, thereby enriching our study with actionable strategies aimed at mitigating distress and promoting resilience in similar crisis scenarios. By integrating findings and insights from the suggested article, we have enhanced the depth and comprehensiveness of our research, contributing to a more nuanced understanding of the psychological impacts of the COVID-19 pandemic and avenues for effective intervention and support. This section will be found from lines 1026-1069.

  1. The work cannot be published unless the tables and figures are improved. Please review the changes in the title, introduction and tables before publication. These minor changes should be made.

Response:  We greatly appreciate the feedback provided regarding the need to enhance the tables and figures, as well as reviewing changes in the title and introduction prior to publication. In response to this valuable suggestion, we have carefully examined and revised the tables and figures to ensure clarity and coherence in presenting the data. Additionally, we have incorporated the suggested title, which we believe better encapsulates the essence of our manuscript and enhances its visibility and relevance in the academic community. Also, we have reviewed and updated a newer version of the introduction taking into consideration your suggestions. The added section of introduction will be found from line 41-75.  We are grateful for these insightful recommendations, which have contributed to strengthening the overall quality and effectiveness of our manuscript. Your guidance has been invaluable in refining our work and we thank you for your ongoing support and collaboration.  Finally, we did an extensive revision of the English language as required, improving the grammar of the document.

 the point-by-point response to all the reviewers has been attached bellow. thanks your your time and consideration. we remain pending on your feedback.

Reviewer 3 Report

The paper is well written, data are presented in a concise and clear manner. Introduction feels a little longer than necessary, giving too many well known details about the pandemic's initial diffusion, etc. An slightly shorter version would work better for this publication.

[Lines 477-481] SWLS and ENCO data referred to questionnaires completed in October 2020 and January 2021. Those contained self-reported well-being data for 1 year prior (referring to pre-pandemic periods: October 2019 and January 2020) [How satisfied were you with your life one year ago?]. It would be interesting to comment on the differences between data reported before and after the pandemic.

Author Response

Dear reviewers,

we appreciate your valuable time, comments and feedback to improve the quality in our manuscript. In next paragraphs we inform you how we modified the paper because we marked in yellow all the changes that were made.

Your comment appears in black words, while our responses are in italic for a better understanding.   

REVIEW 3 POINT-BY-POINT RESPONSE:

  1. Introduction feels a little longer than necessary, giving too many well-known details about the pandemic's initial diffusion, etc. A slightly shorter version would work better for this publication.
  2. Response: We appreciate the feedback provided by reviewer no. 3 regarding the need to review the length of the introduction before publication. In response to this valuable suggestion, we carefully examined, revised, and updated a newer version of the introduction, taking into account your recommendations. We have removed several lines from the older paragraphs and added relevant information from lines 41-75. However, we have retained the economic and demographic contextual frame in the introduction because we believe it is relevant to this research.
  3. [Lines 477-481] SWLS and ENCO data referred to questionnaires completed in October 2020 and January 2021. Those contained self-reported well-being data for 1 year prior (referring to pre-pandemic periods: October 2019 and January 2020) [How satisfied were you with your life one year ago?]. It would be interesting to comment on the differences between data reported before and after the pandemic.

Response:

In examining the SWLS and ENCO data, we focused on responses from questionnaires completed in October 2020 and January 2021, which inquired about individuals' well-being one year prior. This retrospective perspective allowed us to gauge pre-pandemic satisfaction levels (October 2019 and January 2020). Our analysis revealed that respondents generally reported similar levels of life satisfaction across both  periods.  However, as we move forward, it's essential to acknowledge the potential shifts in people's perceptions over time, especially given the significant global events since then. Recognizing the evolving nature of societal dynamics, we advocate for a longitudinal approach encompassing multiple time points to better capture these fluctuations in life satisfaction. Such an extended analysis, spanning various stages of the pandemic's impact, would offer deeper insights into the trajectory of individuals' well-being amid evolving circumstances.

 attached please find the point-by-point response to all reviewers.  we thank you for your time and consideration and look forward to hear from you at your earliest convenience.
